

# A simple coin for a 2d entangled walk

Ahmadullah Zahed[1][*] and Kallol Sen[2][†]

**1** Centre for High Energy Physics, Indian Institute of Science,
C.V. Raman Road, Bangalore 560012, India
**2** ICTP South American Institute for Fundamental Research, IFT-UNESP (*1°* andar),
Rua Dr. Bento Teobaldo Ferraz 271, Bloco 2 - Barra Funda,
01140-070 São Paulo, SP Brazil

[*] ahmadullah@iisc.ac.in , [†] kallolmax@gmail.com

## Abstract

We analyze the effect of a simple coin operator, built out of Bell pairs, in a *2d* Discrete Quantum Random Walk (DQRW) problem. The specific form of the coin enables us to find analytical and closed form solutions to the recursion relations of the DQRW. The coin induces entanglement between the spin and position degrees of freedom, which oscillates with time and reaches a constant value asymptotically. We probe the entangling properties of the coin operator further, by two different measures. First, by integrating over the space of initial tensor product states, we determine the *Entangling Power* of the coin operator. Secondly, we compute the *Generalized Relative Rényi Entropy* between the corresponding density matrices for the entangled state and the initial pure unentangled state. Both the *Entangling Power* and *Generalized Relative Rényi Entropy* behaves similar to the entanglement with time. Finally, in the continuum limit, the specific coin operator reduces the *2d* DQRW into two *1d* massive fermions coupled to synthetic gauge fields, where both the mass term and the gauge fields are built out of the coin parameters.

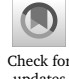
# 1 Introduction

Discrete Quantum Random Walk (DQRW) algorithm has been in study and applications in physics [1–4] and mathematics [5, 6] as well as in the areas of computational intelligence [7, 8], and optimization techniques [9].[1] While the basic features of classical and quantum discrete random walk are essentially the same, one crucial difference between classical and quantum walks is a non-trivial coin operator [11] in the latter case. Most of the random walk explorations are numerical in nature. $1d$ random walks admit analytical solutions, as demonstrated in [12]. However with increasing dimensions, the most general parametric choice for the coin operator also increases and analytical solutions become difficult to obtain. We analyze the DQRW for a two dimensional walk using a special $SU(4)$ coin operator that admits an analytical solution. Our notations follow [11], [13]. We consider a two dimensional walker described by a wave function $|\Psi_w(t)\rangle$

$$|\Psi_w(t)\rangle = \sum_{i=0}^{3} \sum_{m,n=-t}^{t} A_{m,n}^{(i)}(t)\,|i\rangle \times |m,n\rangle \in \mathcal{H}_{spin} \otimes \mathcal{H}_{position}, \qquad (1)$$

that lives in a product space of spin and coordinate degrees of freedom. The product space $\mathcal{H}_{spin} \otimes \mathcal{H}_{position}$ is spanned by a basis of orthonormal vectors $|i\rangle \in \mathcal{H}_{spin}$ and $|m,n\rangle \in \mathcal{H}_{position}$ such that,

$$\langle i|j\rangle = \delta_{ij}, \quad \text{and} \quad \langle m,n|p,q\rangle = \delta_{mp}\delta_{nq}. \qquad (2)$$

The wave function $|\Psi_w(t)\rangle$ is a function of the evolution time $t$ and coordinates $\vec{x} = (m,n)$ such that $-t \le (m,n) \le t$. The wave function evolves discretely with time under the action of the unitary operator $\mathbf{U} = \mathbf{S} \cdot \mathbf{C} \otimes \mathbf{I}_{pos}$, where $\mathbf{I}_{pos}$ is the identity operator in position space, $\mathbf{S}$ is the $2d$ shift operator and $\mathbf{C}$ is the coin operator

$$\mathbf{C}(x,y,z) = \begin{pmatrix} e^{-2i\pi z}\cos(2\pi y) & 0 & 0 & -ie^{-2i\pi z}\sin(2\pi y) \\ 0 & e^{2i\pi z}\cos(2\pi x) & -ie^{2i\pi z}\sin(2\pi x) & 0 \\ 0 & -ie^{2i\pi z}\sin(2\pi x) & e^{2i\pi z}\cos(2\pi x) & 0 \\ -ie^{-2i\pi z}\sin(2\pi y) & 0 & 0 & e^{-2i\pi z}\cos(2\pi y) \end{pmatrix}. \qquad (3)$$

**Main Results:** A $SU(4)$ coin operator has 15 parameters. However, we made a minimal choice involving 3 parameters and it still demonstrates the non-trivial properties of the coin. The simple form (3) renders our $2d$ DQRW algorithm analytically solvable. However (3) induces a non-trivial entanglement between spin and position degrees of freedom. It also exhibits non-trivial entangling properties, which we examine using two different measures. a) Entangling power: where we integrate out the effect of the initial state on the walk. b) Generalized Rényi entropy: where we do a POVM (Positive Valued Operator Measurements) on two density matrix operators between the entangled state and a reference unentangled state. As a bonus, the continuum limit of the algorithm maps to two massive $1+1d$ fermions coupled to gauge fields. We briefly discuss each of these claims below:

---

[1]For a more comprehensive overview of the subject regarding applications and algorithms, please see, [10].

- **Exact Solutions:** Due to the evolution equation, $|\Psi_w(t+1)\rangle = \mathbf{U}|\Psi_w(t)\rangle$, the coefficients $A_{m,n}^{(i)}(t)$ in (1) satisfy a set of discrete recursion relations. Given the specific coin operator (3), we can find closed form analytic solutions for the coefficients $A_{m,n}^{(i)}(t)$, given by,

$$
\begin{aligned}
A_{m,n}^{(0)}(t) &= \delta_{m,n}(1-\delta_{m,-t})e^{-2\pi itz}\left(F_m(y)A_{0,0}^{(0)}(0) + G_m(y)A_{0,0}^{(3)}(0)\right), \\
A_{m,n}^{(1)}(t) &= \delta_{m,-n}(1-\delta_{m,-t})e^{2\pi itz}\left(F_m(x)A_{0,0}^{(1)}(0) + G_m(x)A_{0,0}^{(2)}(0)\right), \\
A_{m,n}^{(2)}(t) &= \delta_{m,-n}(1-\delta_{m,t})e^{2\pi itz}\left(G_{-m}(x)A_{0,0}^{(1)}(0) + F_{-m}(x)A_{0,0}^{(2)}(0)\right), \\
A_{m,n}^{(3)}(t) &= \delta_{m,n}(1-\delta_{m,t})e^{-2\pi itz}\left(G_{-m}(y)A_{0,0}^{(0)}(0) + F_{-m}(y)A_{0,0}^{(3)}(0)\right),
\end{aligned}
\tag{4}
$$

$$
\begin{aligned}
F_m(x) =& \frac{(-1)^{\frac{t-m}{2}}\sin^2(2\pi x)\Gamma\left(\frac{m+t+2}{2}\right)\cos^m(2\pi x)}{\Gamma(m+1)\Gamma\left(\frac{-m+t+2}{2}\right)} \\
& \times {}_2F_1\left(\frac{m-t+2}{2}, \frac{m+t+2}{2}; m+1; \cos^2(2\pi x)\right), \\
G_m(x) =& -i\sin(2\pi x)\cos^{t-1}(2\pi x)\,{}_2F_1\left(\frac{2-m-t}{2}, \frac{m-t}{2}; 1; -\tan^2(2\pi x)\right).
\end{aligned}
\tag{5}
$$

- **Probablity and Entanglement:** The probability distribution as a function of the coordinates is given by,

$$
P_{m,n}(t) = \sum_{i=0}^{3}|A_{m,n}^{(i)}(t)|^2, \qquad \sum_{m,n=-t}^{t}P_{m,n}(t) = 1.
\tag{6}
$$

The density matrix for the walker,

$$
\rho(t) = |\Psi_w(t)\rangle\langle\Psi_w(t)|, \qquad \widetilde{\rho}(t) = \mathrm{tr}_{pos}|\Psi_w(t)\rangle\langle\Psi_w(t)|.
\tag{7}
$$

$E(t) = -\widetilde{\rho}(t)\log\widetilde{\rho}(t)$ is the entanglement entropy between $\mathcal{H}_{spin}$ and $\mathcal{H}_{position}$ degrees of freedom. If we consider the spin Hilbert space to be $\mathcal{H}_{spin} = \mathcal{H}_A \otimes \mathcal{H}_B$, where $A, B$ are the subsystems in the spin space, then the entanglement between $A$ and $B$ as a function of the grid is obtained from,

$$
E_{m,n}(t) = -\rho_{m,n}^A(t)\log\rho_{m,n}^A(t), \quad \text{where} \quad \rho_{m,n}^A(t) = \mathrm{tr}_B\frac{\langle m,n|\rho^{AB}(t)|m,n\rangle}{\langle\Psi_w(t)|m,n\rangle\langle m,n|\Psi_w(t)\rangle}.
\tag{8}
$$

The function is normalized so that $\mathrm{tr}\,\rho_{m,n}^A(t) = 1$. The entanglement between the position and spin dofs, is given by, $E_C(t) = -\widetilde{\rho}(t)\log\widetilde{\rho}(t)$. The asymptotic entanglement (for $x = y = 1/8$ and $z = 1/10$) is given by,

$$
E_{1/8,1/8,1/10}(t) = 0.693156 - \frac{\cos^2(\pi t/2)}{4t^2} + \frac{\sin^2(\pi t/4 + \pi/8)}{2t^{5/2}} + \ldots,
\tag{9}
$$

while the same functional behavior is also evident for other parametric choices, as demonstrated in (36).

- **Entangling Power:** The *entangling power* [14, 15] of the coin on the random walk is defined as,

$$
\mathcal{E}_C(t) = \frac{1}{V}\int_{\psi_{initial}} dV\sqrt{g}\left(1 - \mathrm{tr}\,\widetilde{\rho}(t)^2\right).
\tag{10}
$$

where $\rho_{\mathbf{C}}(t)$ is the reduced density matrix after tracing over position. For example, the asymptotic entangling power of the coin operator,

$$\mathcal{E}_{\mathbf{C}}(t) = 0.671914 - \frac{0.0318321 \sin\left(\frac{\pi}{8} - \frac{\pi t}{2}\right)}{\sqrt[4]{t}} + \dots, \quad x = y = 1/8, \quad z = 1/10, \quad (11)$$

oscillates with time and approaches a constant as $t \to \infty$. The entanglement follows the exact same functional behaviour for other parametric choices as well, albeit with different numerical coefficients as demonstrated in (45).

- **Generalized Rényi Entropies:** We compute and compare two distinct definitions [16–19], *viz. $\alpha$−Sandwiched Renyi Divergence* (SRD) and *$\alpha$−Relative Renyi Entropy* (RRE). Given two operators $\rho$ and $\sigma$, the $\alpha$−SRD and $\alpha$−RRE are given by,

$$D_{\alpha-RRE} = \frac{1}{\alpha-1}\log\frac{\mathrm{tr}\,\rho^{\alpha}\sigma^{1-\alpha}}{\mathrm{tr}\,\rho}, \quad \widetilde{D}_{\alpha-SRD} = \frac{1}{\alpha-1}\log\frac{\mathrm{tr}\,\sigma^{(1-\alpha)/2\alpha}\rho\sigma^{(1-\alpha)/2\alpha}}{\mathrm{tr}\,\rho}, \quad (12)$$

for $\rho \not\perp \sigma$ and $\alpha \in [0,1)$. For our case, $\rho = \widetilde{\rho}(t)$ denotes the density operator for the final entangled mixed state $|\Psi_w(t)\rangle$ and $\sigma$ is a reference operator corresponding to a pure state. For our case, the asymptotic form of the $\alpha$−SRD and $\alpha$−RRE for large $t$ is given by,

$$D_{1/4-SRD} = 0.379594 + \frac{0.106996 \sin\left(\frac{\pi t}{4} + \frac{\pi}{16}\right)}{t^{3/2}} -$$
$$\frac{0.157844 \cos\left(\frac{\pi t}{2} + \frac{\pi}{16}\right)}{\sqrt{t}} + \dots, \quad x = y = 1/8, \quad (13)$$

$$D_{1/4-RRE} = 0.389889 + \frac{0.0955922 \sin\left(\frac{\pi t}{4} + \frac{\pi}{16}\right)}{t^{3/2}} -$$
$$\frac{0.179794 \cos\left(\frac{\pi t}{2} + \frac{\pi}{16}\right)}{\sqrt{t}} + \dots, \quad x = y = 1/8. \quad (14)$$

We compute similar expressions for asymptotic forms of the $\alpha$−SRD and $\alpha$−RRE for other parametric choices of the coin operator and for $\alpha = 1/2, 3/4$ in (53)-(57). The functional forms of the asymptotic expansions for large $t$ are similar with different numerical coefficients depending on the coin parameters and $\alpha$.

- **Continuum limit:** In the continuum limit [1, 20, 21], the recursion relations reduce to the Dirac equation in $1+1$ dimensions,

$$(\gamma^{\mu} D_{\mu}^{\pm} - \mathcal{M}_{\pm})\psi_{\pm} = 0, \quad (15)$$

of two massive fermions $(\psi_+, \psi_-)$ coupled to gauge fields. $D_{\mu}^{\pm} = \partial_{\mu} - iA_{\mu}^{\pm}$ where $A_{\mu}^{\pm} = (V, 0, 0, 0)$. The Random walk wave function in the continuum limit is related to the fermions, by,

$$|\Psi_w(t,\vec{x})\rangle = M \cdot (|\uparrow\rangle \otimes |\psi_+\rangle + |\downarrow\rangle \otimes |\psi_-\rangle), \quad \text{where} \quad M = \begin{pmatrix} 1 & 0 & 0 & 0 \\ 0 & 0 & 1 & 0 \\ 0 & 0 & 0 & 1 \\ 0 & 1 & 0 & 0 \end{pmatrix}. \quad (16)$$

The spectrum of such a two particle system, is given by,

$$E(p_1, p_2) = 2\alpha \pm \sqrt{p_1^2 + \theta_1^2} \pm \sqrt{p_2^2 + \theta_2^2}. \quad (17)$$

Positive energy conditions, imply that $\theta_i \leq V_i$.

The remainder of the work will be organized as follows. In section 2, we give the mathematical setup of the problem. We clarify the notations and construct the analytical solutions to the recursion relations. In section 3, we consider the probability and entanglement distributions over the two dimensional grid. We also provide the entanglement of the spin and position degrees of freedom as a function of time. In section 4 we discuss the entangling power of the coin operator on the walk as a function of time and coin parameters. The function approaches a constant at large times similar to the entanglement. Section 5 discusses and compares the distinct definitions of the *Quantum Dynamical Entropy* from the random walk perspective. For very simple reference matrices, we can obtain exact analytical solutions for both, while numerical results suffice for generic reference states. Section 6 discusses the continuum limit of the walk, which reduces to two one dimensional fermions coupled with gauge fields. We end the work with discussions of future directions in section 7.

## 2 Setup: Coin operator

The wave function for the two dimensional Quantum Discrete Random Walk, can be written in a form,

$$|\Psi_w(t)\rangle = \sum_{i=0}^{3} \sum_{m,n=-t}^{t} A_{m,n}^{(i)}(t) |i\rangle \otimes |m,n\rangle. \tag{18}$$

The wave function $|\Psi_w(t)\rangle \in \mathcal{H}_{spin} \otimes \mathcal{H}_{position}$ where $|i\rangle \in \mathcal{H}_{spin}$ is the spin Hilbert space and $|m,n\rangle \in \mathcal{H}_{position}$ is the coordinate Hilbert space. $|i\rangle \in \mathcal{H}_{spin}$ and $|m,n\rangle \in \mathcal{H}_{position}$ are set of orthornormal vectors, such that,

$$\langle i|j\rangle = \delta_{ij} \text{ and } \langle m,n|p,q\rangle = \delta_{mp}\delta_{nq}. \tag{19}$$

The position grid is a finite size system $\mathfrak{dim}(m,n) = (2t+1) \times (2t+1)$. The unitary evolution of the random walk is governed by the equation,

$$|\Psi_w(t)\rangle = \mathbf{U}|\Psi_w(t-1)\rangle, \quad \mathbf{U} = \mathbf{S} \cdot \mathbf{C} \otimes \mathbb{I}_{pos}, \tag{20}$$

where $\mathbf{U}$ is a unitary operator, and

$$\mathbf{S} = \sum_{i=0}^{3} \sum_{\vec{x}} |i\rangle\langle i| \otimes |\vec{x} + \alpha_i\rangle\langle\vec{x}|, \tag{21}$$

with $\alpha_0 = (1,1)$, $\alpha_1 = (1,-1)$, $\alpha_2 = -\alpha_1$ and $\alpha_3 = -\alpha_0$, is the *Shift Operator* and we choose a specific coin operator $\mathbf{C}$ for our purposes, built out of Bell pairs,

$$\mathbf{C} = \sum_{k=1}^{4} e^{i\lambda_k} |\Phi_k\rangle\langle\Phi_k|, \quad \sum_{k=1}^{4} \lambda_k = 0, \tag{22}$$

for $\lambda_1 + \lambda_2 = -4\pi z$, $\lambda_1 - \lambda_2 = -4\pi y$, $\lambda_3 - \lambda_4 = -4\pi x$ and where,

$$|\Phi_{1,2}\rangle = \frac{|0\rangle \pm |3\rangle}{\sqrt{2}}, \quad |\Phi_{3,4}\rangle = \frac{|1\rangle \pm |2\rangle}{\sqrt{2}}. \tag{23}$$

Explicitly,

$$\mathbf{C}(x,y,z) = \begin{pmatrix} e^{-2i\pi z}\cos(2\pi y) & 0 & 0 & -ie^{-2i\pi z}\sin(2\pi y) \\ 0 & e^{2i\pi z}\cos(2\pi x) & -ie^{2i\pi z}\sin(2\pi x) & 0 \\ 0 & -ie^{2i\pi z}\sin(2\pi x) & e^{2i\pi z}\cos(2\pi x) & 0 \\ -ie^{-2i\pi z}\sin(2\pi y) & 0 & 0 & e^{-2i\pi z}\cos(2\pi y) \end{pmatrix}. \tag{24}$$

From the explicit form of the unitary operator **U** and the evolution equation (20), the recursion relation for the coefficients $A^{(i)}_{m,n}(t)$ reads,

$$A^{(0)}_{m,n}(t) = e^{-2i\pi z}\cos(2\pi y)A^{(0)}_{m-1,n-1}(t-1) - ie^{-2i\pi z}\sin(2\pi y)A^{(3)}_{m-1,n-1}(t-1),$$
$$2-t \leq m,n \leq t,$$
$$A^{(1)}_{m,n}(t) = e^{2i\pi z}\cos(2\pi x)A^{(1)}_{m-1,n+1}(t-1) - ie^{2i\pi z}\sin(2\pi x)A^{(2)}_{m-1,n+1}(t-1),$$
$$2-t \leq m \leq t, -t \leq n \leq t-2,$$
$$A^{(2)}_{m,n}(t) = -ie^{2i\pi z}\sin(2\pi x)A^{(1)}_{m+1,n-1}(t-1) + e^{2i\pi z}\cos(2\pi x)A^{(2)}_{m+1,n-1}(t-1),$$
$$-t \leq m \leq t-2, 2-t \leq n \leq t,$$
$$A^{(3)}_{m,n}(t) = -ie^{-2i\pi z}\sin(2\pi y)A^{(0)}_{m+1,n+1}(t-1) + e^{-2i\pi z}\cos(2\pi y)A^{(3)}_{m+1,n+1}(t-1),$$
$$-t \leq m,n \leq t-2. \tag{25}$$

Due to the special nature of the coin operator, the recursion relations decouple in the sense that $(A^{(0)}_{m,n}(t), A^{(3)}_{m,n}(t))$ decouple from $(A^{(1)}_{m,n}(t), A^{(2)}_{m,n}(t))$. The recursion relations can be solved exactly and we can write the analytical forms of these coefficients in the form,

$$A^{(0)}_{m,n}(t) = \delta_{m,n}(1-\delta_{m,-t})e^{-2\pi itz}\left(F_m(y)A^{(0)}_{0,0}(0) + G_m(y)A^{(3)}_{0,0}(0)\right),$$
$$A^{(3)}_{m,n}(t) = \delta_{m,n}(1-\delta_{m,t})e^{-2\pi itz}\left(G_{-m}(y)A^{(0)}_{0,0}(0) + F_{-m}(y)A^{(3)}_{0,0}(0)\right),$$
$$A^{(1)}_{m,n}(t) = \delta_{m,-n}(1-\delta_{m,-t})e^{2\pi itz}\left(F_m(x)A^{(1)}_{0,0}(0) + G_m(x)A^{(2)}_{0,0}(0)\right),$$
$$A^{(2)}_{m,n}(t) = \delta_{m,-n}(1-\delta_{m,t})e^{2\pi itz}\left(G_{-m}(x)A^{(1)}_{0,0}(0) + F_{-m}(x)A^{(2)}_{0,0}(0)\right), \tag{26}$$

with the initial wave function at the origin,

$$|\Psi(0)\rangle = \begin{pmatrix} A^{(0)}_{0,0}(0) \\ A^{(1)}_{0,0}(0) \\ A^{(2)}_{0,0}(0) \\ A^{(3)}_{0,0}(0) \end{pmatrix} \otimes |0,0\rangle. \tag{27}$$

The functions $F_m$ and $G_m$ are,

$$F_m(x) = \frac{(-1)^{\frac{t-m}{2}}\sin^2(2\pi x)\Gamma\left(\frac{m+t+2}{2}\right)\cos^m(2\pi x)}{\Gamma(m+1)\Gamma\left(\frac{-m+t+2}{2}\right)}$$
$$\times {}_2F_1\left(\frac{m-t+2}{2}, \frac{m+t+2}{2}; m+1; \cos^2(2\pi x)\right), \tag{28}$$
$$G_m(x) = -i\sin(2\pi x)\cos^{t-1}(2\pi x) {}_2F_1\left(\frac{2-m-t}{2}, \frac{m-t}{2}; 1; -\tan^2(2\pi x)\right).$$

For the rest of the paper, we will use specific choice for the coin parameters. These are,

$$\mathbf{C}(1/8, 1/8, 1/10), \quad \mathbf{C}(1/8, 1/12, 1/10) \quad \text{and} \quad \mathbf{C}(1/6, 1/8, 1/10). \tag{29}$$

## 3 Probability and Entanglement

The probability distribution is a function of the $2D$ grid, and is given by,

$$P_{m,n}(t) = \langle m,n|\Psi_w(t)\rangle\langle\Psi_w(t)|m,n\rangle = \sum_{i=0}^{3}\left|A^{(i)}_{m,n}(t)\right|^2, \quad \sum_{m,n=-t}^{t}P_{m,n}(t) = 1. \tag{30}$$

The density operator given by,

$$\rho(t) = |\Psi_w(t)\rangle\langle\Psi_w(t)|, \tag{31}$$

lives in $(8t+4) \times (8t+4)$ dimensions, from which we can define the density matrix on the 2D grid to be,

$$\rho_{m,n}(t) = \frac{\langle m,n|\rho(t)|m,n\rangle}{|\langle m,n|\Psi_w(t)\rangle|^2}. \tag{32}$$

An alternative way to look at this expression is the following. As a small digression, if we consider the spin Hilbert space $\mathcal{H}_{spin}(= \mathcal{H}_A \otimes \mathcal{H}_B)$ to be composed of two subsystems $A$ and $B$ (meaning $\rho_{m,n} = \rho_{m,n}^{AB}$), then tracing over either subsystem, provides a definition of the entanglement between $A$ and $B$

$$E_{m,n}(t) = -\text{tr } \rho_{m,n}^A(t) \log \rho_{m,n}^A(t), \quad \rho_{m,n}^A(t) = \text{tr }_B \rho_{m,n}^{AB}(t), \tag{33}$$

as a function of the grid. We plot the probability and entanglement distribution over the two dimensional grid in figure 1 for the coin parameters $x = 1/6, y = 1/8$ and $z = 1/10$. The plots for other parametric choices are similar. Note that the probability and entanglement distribution on the grid is non-vanishing only along the diagonals, which reflects the decoupled recursion relations in (25). On the other hand, tracing over the entire grid, gives a reduced density matrix,

$$\widetilde{\rho}(t) = \text{tr }_{pos}\rho(t), \quad E(t) = -\widetilde{\rho}(t)\log\widetilde{\rho}(t), \tag{34}$$

that defines the entanglement between the spin and position degrees of freedom. Below we plot the probability and the entanglement distribution for a tensor product initial state,

$$|\Psi_w(0)\rangle = |0\rangle \otimes \frac{|0\rangle + |1\rangle}{\sqrt{2}} \otimes |0,0\rangle. \tag{35}$$

The entanglement of the spin and position degrees of freedom, as given by (34), is a function of the time, as we show in figure 2. For the choices of coin parameters, we can fit the asymptotic entanglement to the forms,

$$E_{1/6,1/8,1/10}(t) = 0.695062 - 0.042797\frac{\cos^2(\pi t/4 + \pi/8)}{\sqrt{t}} \tag{36}$$

$$- 0.0567782\frac{\sin^2(\pi t/3 + \pi/6)}{\sqrt{t}} + \dots, \tag{37}$$

$$E_{1/8,1/8,1/10}(t) = 0.693156 - \frac{\cos^2(\pi t/2)}{4t^2} + \frac{\sin^2(\pi t/4 + \pi/8)}{2t^{5/2}} + \dots, \tag{38}$$

$$E_{1/8,1/12,1/10}(t) = 0.69212 - 0.0207781\frac{\cos^2(\pi t/6 + \pi/12)}{\sqrt{t}} \tag{39}$$

$$- 0.046477\frac{\sin^2(\pi t/4 + \pi/8)}{\sqrt{t}} + \dots, \tag{40}$$

where $\dots$ represent sub-leading terms. The asymptotic entanglement for the entangled coin is below the minimal value for entanglement for Grover's and Kempe's coin for $2d$ random walk [22], [23].

A general form of the asymptotic form of entanglement for our choice of coin can be written as,

$$E_{x,y,z}(t) = A_0(x,y,z) + \sum_{i,m\geq 0} \rho_i \frac{\sin^2(\pi t/a_i + \gamma_i)}{t^m} + \sum_{i,n\geq 0} \sigma_i \frac{\cos^2(\pi t/b_i + \delta_i)}{t^n}, \tag{41}$$

where we have the first leading coefficients in (36).

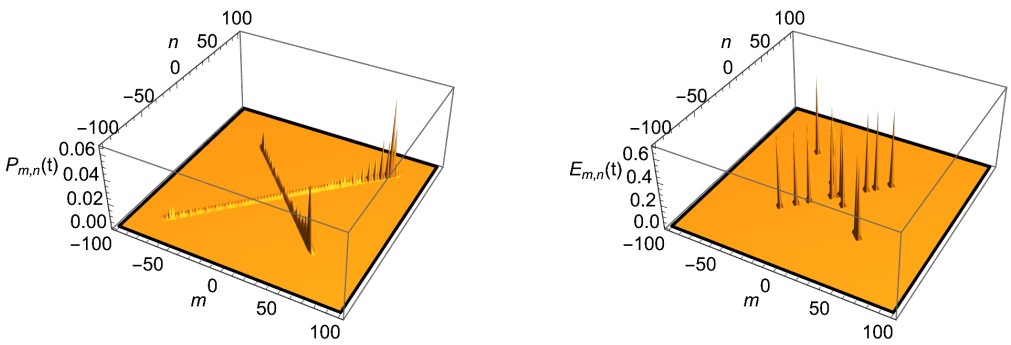

Figure 1: $P_{m,n}(t)$ and $E_{m,n}(t)$ for $t = 100$ for $\mathbf{C}(1/6, 1/8, 1/10)$ for the initial tensor product state in (35).

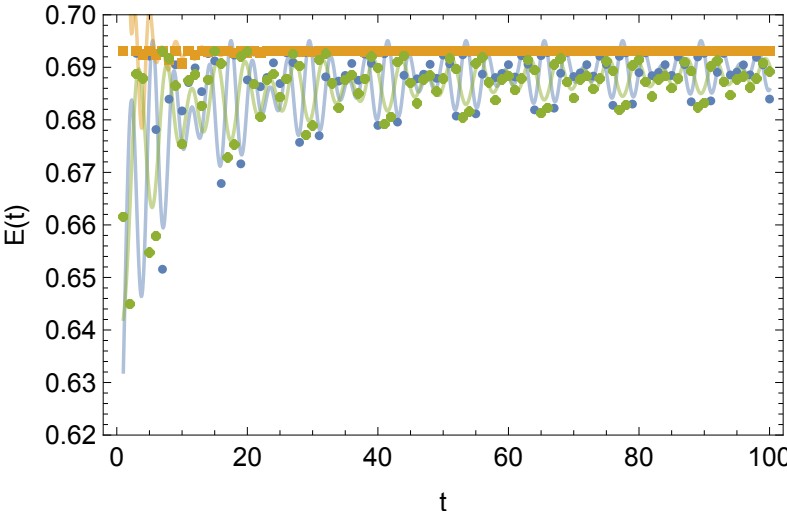

Figure 2: $E(t)$ vs. $t$ for coins $C(1/6, 1/8, 1/10)$ (blue), $C(1/8, 1/8, 1/10)$ (orange) and $C(1/8, 1/12, 1/10)$ (green).

## 4 Entangling power

To probe the entangling properties of the coin operator in (24) further, we analyze its *Entangling power* [14], [15], [24], [25,26], [27], which describes the capacity of the coin operator to produce an entangled state from the initial tensor-product states. For concreteness, consider the initial state,

$$|\Psi_w(0)\rangle = (|\psi_1\rangle \otimes |\psi_2\rangle) \otimes |0, 0\rangle, \tag{42}$$

where,

$$|\psi_i\rangle = \cos\frac{\theta_i}{2}|0\rangle + e^{i\alpha_i}\sin\frac{\theta_i}{2}|1\rangle, \tag{43}$$

implying that $|\Psi(0)\rangle \in \mathbb{CP}_1 \otimes \mathbb{CP}_1$.

$$\mathcal{E}_{\mathbf{C}}(t) = \frac{1}{16\pi^2}\int_{\mathcal{M}} d\theta_1 d\theta_2 d\alpha_1 d\alpha_2 \, \sin\theta_1 \sin\theta_2 \left(1 - \operatorname{tr}\widetilde{\rho}(t)^2\right) \tag{44}$$

gives the *entangling power* or equivalently "*the capacity to entangle*" for the coin operator in the random walk. The integral is over the entire manifold of tensor product states,

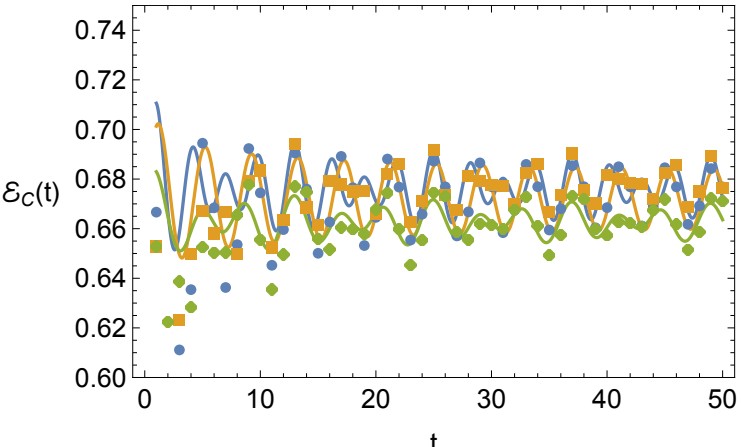

Figure 3: The plot shows the fitted functional forms (solid lines) against the data obtained for $\mathcal{E}_C(t)$ vs. $t$ for up to $t = 50$ for three parametric choices $\mathbf{C}(1/8, 1/8, 1/10)$ (blue), $\mathbf{C}(1/6, 1/8, 1/10)$ (orange) and $\mathbf{C}(1/8, 1/12, 1/10)$ (green).

$\mathcal{M} = \mathbb{CP}_1 \otimes \mathbb{CP}_1$. The integral limits are $0 \le (\theta_1, \theta_2) \le \pi$ and $0 \le (\alpha_1, \alpha_2) \le 2\pi$. In Fig (3), we explore the entangling power $\mathcal{E}_C(t)$ for three parametric choices of the coin operator.

The functional forms for the asymptotic ($t \gg 1$) entangling power (for fixed $z = 0.1$), are given by,

$$
\mathcal{E}_{\mathbf{C}}(t) = 
\begin{cases}
0.671914 - \dfrac{0.0318321 \sin\left(\frac{\pi}{8} - \frac{\pi t}{2}\right)}{\sqrt[4]{t}} + \dots, & x = y = 1/8, \\[3ex]
0.676256 + \dfrac{0.0123798 \sin\left(\frac{\pi t}{2}\right)}{\sqrt[4]{t}} - \dfrac{0.0219393 \sin\left(\frac{\pi}{6} - \frac{2\pi t}{3}\right)}{\sqrt[4]{t}} + \dots, & x = 1/6, y = 1/8, \\[3ex]
0.662477 + \dfrac{0.00779714 \sin\left(\frac{\pi t}{3} + \frac{\pi}{12}\right)}{\sqrt[4]{t}} + \dfrac{0.0131973 \sin\left(\frac{\pi t}{2} + \frac{\pi}{16}\right)}{\sqrt[4]{t}} + \dots, & x = 1/8, y = 1/12.
\end{cases}
$$
(45)

This suggests that the leading general asymptotic form of the *Entangling Power* as a function of time and the coin parameters will take the form,

$$
\mathcal{E}_{\mathbf{C}}(t) = \mathcal{A}_{\mathbf{C}} + \sum_{i, m \ge 0} \rho_i \frac{\sin^2(\pi t / a_i + \gamma_i)}{t^{m + 1/4}} + \sum_{i, n \ge 0} \sigma_i \frac{\cos^2(\pi t / b_i + \delta_i)}{t^{n + 1/4}} + \dots,
$$
(46)

where $\mathcal{A}_C$ is time independent and function of the coin parameters. The parameters, $\rho_i, \sigma_i, a_i, b_i, \gamma_i, \delta_i$ depend on the coin parameters.

## 5 Generalized relative Rényi entropy

In addition to the *Entangling Power*, a second measure to elucidate the entangling properties of the coin, is using the definitions of *Generalized Relative Rényi Entropy*. Since the walk algorithm describes a quantum process, it is necessary to use observables which capture the quantum nature of the algorithm. Following [16] [17], [18, 19], we compute two definitions from our quantum walk perspective *viz.* $\alpha-$*Sandwiched Rényi Divergence* and $\alpha-$*Relative Rényi Entropy*. These are defined as follows. Given two positive operators $\rho$ and $\sigma$ such that $\rho \not\perp \sigma$, we can define,

$$
D_{\alpha-SRD}(\rho, \sigma) = \frac{1}{\alpha - 1} \log \frac{\operatorname{tr} \sigma^{\frac{1 - \alpha}{2\alpha}} \rho \sigma^{\frac{1 - \alpha}{2\alpha}}}{\operatorname{tr} \rho}, \qquad D_{\alpha-RRE}(\rho, \sigma) = \frac{1}{\alpha - 1} \log \frac{\operatorname{tr} \rho^{\alpha} \sigma^{1 - \alpha}}{\operatorname{tr} \rho}, \quad (47)
$$

for $\alpha \in [0, 1)$. For any quantum operation $\Lambda : \rho \to \Lambda\rho$, the definitions satisfy,

$$D_\alpha(\rho, \sigma) \geq D_\alpha(\Lambda(\rho), \Lambda(\sigma)) . \tag{48}$$

Note that,

$$[\rho, \sigma] = 0 \quad \Rightarrow \quad D_{\alpha - SRD} = D_{\alpha - RRE} = D_\alpha^{\text{cl}}, \tag{49}$$

where,

$$D_\alpha^{\text{cl}}(p, q) = \frac{1}{\alpha - 1} \log \frac{\sum_{x \in \mathcal{X}} p(x)^\alpha q(x)^{1-\alpha}}{\sum_{x \in \mathcal{X}} p(x)}, \quad \text{supp } p(x) \subseteq \text{supp } p(x) \wedge \alpha \neq 1, \tag{50}$$

is the classical Relative Rényi Entropy for two positive probability distributions $p(x)$ and $q(x)$ over a set $x \in \mathfrak{X}$, such that supp $p(x) \subseteq$ supp $q(x)$.[2] For computational purposes, we take the following definitions of $\rho$ and $\sigma$. We take,

$$\rho(t) = \text{tr }_{pos} |\Psi_w(t)\rangle\langle\Psi_w(t)|, \quad \text{and} \quad \sigma = |\Psi_w(0)\rangle\langle\Psi_w(0)| . \tag{51}$$

Note that $\rho(t)$ is the density operator for a mixed state (from tracing over the position coordinates) while $\sigma$ is the density operator for a pure state. Both satisfy tr $\rho = $ tr $\sigma = 1$. We will choose specific initial conditions for the entropic measures,[3]

$$|\Psi_w(0)\rangle = \frac{(1, i, 0, 0)}{\sqrt{2}} \otimes |0, 0\rangle . \tag{52}$$

$\rho(t)$ can be computed analytically using the solutions in (26). We plots $D_{\alpha - SRD}$ and $D_{\alpha - RRE}$ as functions of time ($t$), in figure 4 for three parameter choices of the coin operator in (29).

[t] From figure 4, we can fit the leading asymptotic form of $D_{\alpha - SRD}$ for large $t$, to the following functional form (we fix $z = 0.1$ for all computations),

$$D_{1/4 - SRD} = \begin{cases} 0.379594 + \frac{0.106996 \sin\left(\frac{\pi t}{4} + \frac{\pi}{16}\right)}{t^{3/2}} - \frac{0.157844 \cos\left(\frac{\pi t}{2} + \frac{\pi}{16}\right)}{\sqrt{t}} + \dots, & x = y = 1/8, \\ 0.352781 - \frac{0.0568579 \cos\left(\frac{\pi t}{3} + \frac{\pi}{12}\right)}{\sqrt{t}} - \frac{0.0932246 \cos\left(\frac{\pi t}{2} + \frac{\pi}{8}\right)}{\sqrt{t}} + \dots, & x = 1/8, y = 1/12, \\ 0.401346 - \frac{0.113507 \sin\left(\frac{\pi}{6} - \frac{\pi t}{2}\right)}{\sqrt{t}} - \frac{0.0769188 \cos\left(\frac{2\pi t}{3} + \frac{\pi}{4}\right)}{\sqrt{t}} + \dots, & x = 1/6, y = 1/8. \end{cases} \tag{53}$$

The same functional basis exists for $\alpha = 1/2$ and $\alpha = 3/4$ as well, such that,

$$D_{1/2 - SRD} \sim 3 D_{1/4 - SRD} + \dots \quad \text{and} \quad D_{3/4 - SRD} \sim 9 D_{1/4 - SRD} + \dots \tag{54}$$

Similarly for the leading asymptotic form of $\alpha - RRE$ at large $t$, we find the functional form,

$$D_{1/4 - RRE} = \begin{cases} 0.389889 + \frac{0.0955922 \sin\left(\frac{\pi t}{4} + \frac{\pi}{16}\right)}{t^{3/2}} - \frac{0.179794 \cos\left(\frac{\pi t}{2} + \frac{\pi}{16}\right)}{\sqrt{t}} + \dots, & x = y = 1/8, \\ 0.363284 - \frac{0.0645552 \cos\left(\frac{\pi t}{3} + \frac{\pi}{12}\right)}{\sqrt{t}} - \frac{0.10113 \cos\left(\frac{\pi t}{2} + \frac{\pi}{8}\right)}{\sqrt{t}} + \dots, & x = 1/8, y = 1/12, \\ 0.411921 - \frac{0.118399 \sin\left(\frac{\pi}{6} - \frac{\pi t}{2}\right)}{\sqrt{t}} - \frac{0.0762127 \cos\left(\frac{2\pi t}{3} + \frac{\pi}{4}\right)}{\sqrt{t}} + \dots, & x = 1/6, y = 1/8, \end{cases} \tag{55}$$

$$D_{1/2 - RRE} = \begin{cases} 1.15822 + \frac{0.300902 \sin\left(\frac{\pi t}{4} + \frac{\pi}{16}\right)}{t^{3/2}} - \frac{0.51888 \cos\left(\frac{\pi t}{2} + \frac{\pi}{16}\right)}{\sqrt{t}} + \dots, & x = y = 1/8, \\ 1.07782 - \frac{0.185198 \cos\left(\frac{\pi t}{3} + \frac{\pi}{12}\right)}{\sqrt{t}} - \frac{0.296234 \cos\left(\frac{\pi t}{2} + \frac{\pi}{8}\right)}{\sqrt{t}} + \dots, & x = 1/8, y = 1/12, \\ 1.22452 - \frac{0.353662 \sin\left(\frac{\pi}{6} - \frac{\pi t}{2}\right)}{\sqrt{t}} - \frac{0.228691 \cos\left(\frac{2\pi t}{3} + \frac{\pi}{4}\right)}{\sqrt{t}} + \dots, & x = 1/6, y = 1/8, \end{cases} \tag{56}$$

---

[2]This implies that $p(x) = 0$ whenever $q(x) = 0$.

[3]Note that this choices are by no means unique and other initial conditions are equally valid. We however stress that the functional forms for the asymptotic behavior of the entropy functions as $t \to \infty$ will not be affected by the specific forms of the initial conditions.

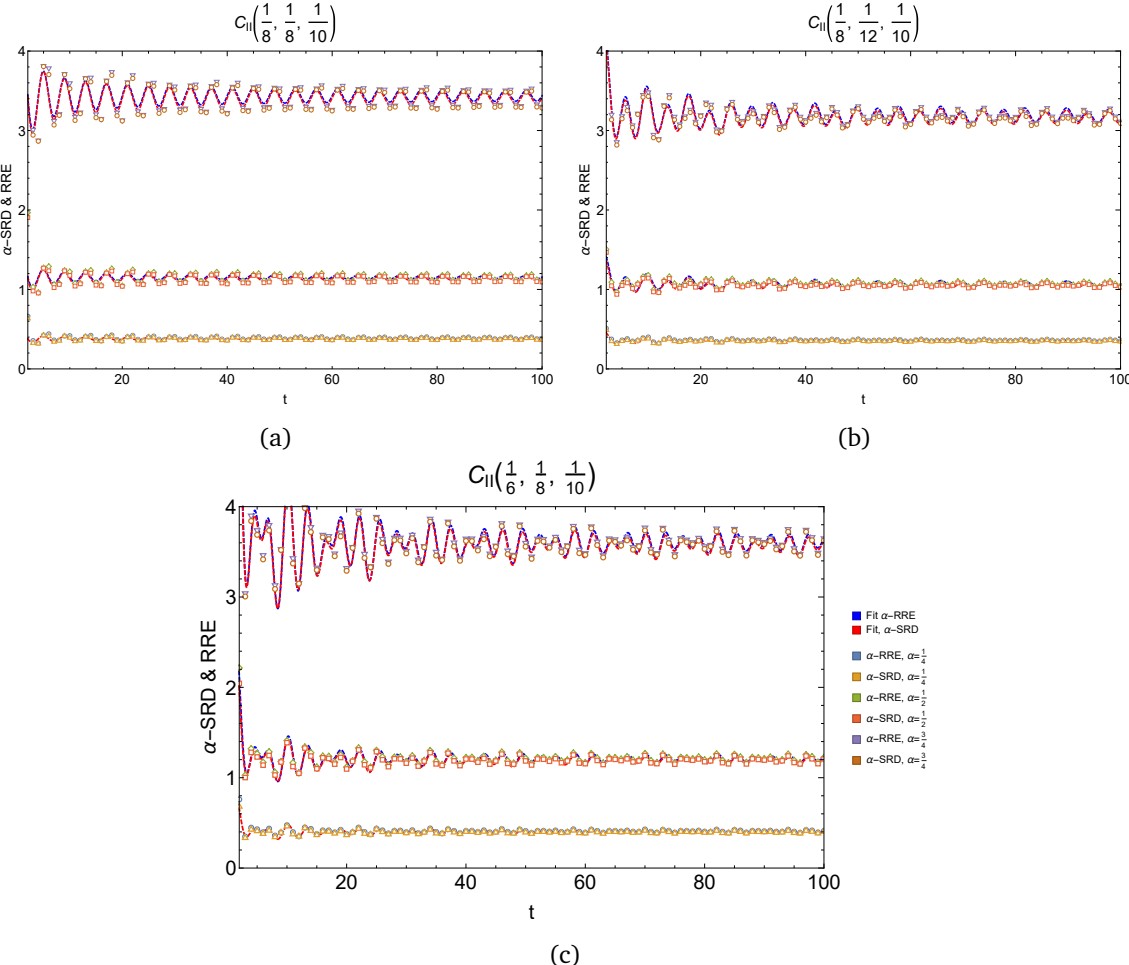

Figure 4: The plot shows the fitted functional forms (given by the dashed lines) against the data obtained for $\alpha-$SRD and $\alpha-$RRE vs. $t$ for up to $t = 100$ for the chosen coin parameters labelling the plots. We choose $\alpha = 1/4, 1/2, 3/4$.

and,

$$
D_{3/4-RRE} = \begin{cases} 3.44391 + \dfrac{0.937397\sin\left(\frac{\pi t}{4}+\frac{\pi}{16}\right)}{t^{3/2}} - \dfrac{1.49066\cos\left(\frac{\pi t}{2}+\frac{\pi}{16}\right)}{\sqrt{t}} + \ldots, & x = y = 1/8, \\[2ex] 3.20219 - \dfrac{0.532561\cos\left(\frac{\pi t}{3}+\frac{\pi}{12}\right)}{\sqrt{t}} - \dfrac{0.864633\cos\left(\frac{\pi t}{2}+\frac{\pi}{8}\right)}{\sqrt{t}} + \ldots, & x = 1/8, y = 1/12, \\[2ex] 3.64189 - \dfrac{1.04617\sin\left(\frac{\pi}{6}-\frac{\pi t}{2}\right)}{\sqrt{t}} - \dfrac{0.68791\cos\left(\frac{2\pi t}{3}+\frac{\pi}{4}\right)}{\sqrt{t}} + \ldots, & x = 1/6, y = 1/8, \end{cases}
\tag{57}
$$

Given the asymptotic forms, we can infer a general asymptotic form for both $\alpha - SRD$ and $\alpha - RRE$, given by,

$$
F(\alpha, \mathbf{C}) = A_{\alpha,\mathbf{C}} + \sum_{i,m} \rho_i \frac{\cos(4\pi t/a_i + \gamma_i)}{t^{m+1/2}} + \sum_{i,n} \sigma_i \frac{\sin(4\pi t/b_i + \delta_i)}{t^{n+1/2}}.
\tag{58}
$$

## 6 Continuum limit

For our choice of coin (24), the continuum limit of the quantum discrete random walk gives rise to synthetic gauge fields [20, 21]. To start with, we replace the discrete differences with

derivatives,

$$|\Psi_w(t+\Delta t,\vec{x})\rangle - |\Psi_w(t,\vec{x})\rangle = \Delta t\,\partial_t|\Psi_w(t,\vec{x})\rangle\,,\quad |\Psi_w(t,\vec{x}+\Delta\vec{x})\rangle - |\Psi_w(t,\vec{x})\rangle = \Delta\vec{x}\cdot\partial_{\vec{x}}|\Psi_w(t,\vec{x})\rangle\,.$$
(59)

Using $|\psi_w(t+\Delta t,\vec{x})\rangle = \mathbf{U}|\Psi_w(t,\vec{x})\rangle$, we can also write the analog continuum version of (20),

$$\partial_t|\Psi_w(t,\vec{x})\rangle = \frac{\mathbf{U}-\mathbf{I}}{\Delta t}|\Psi_w(t,\vec{x})\rangle\,,\qquad \mathbf{U}=\mathbf{S}\cdot\mathbf{C}\,.$$
(60)

In the continuum limit (for $\Delta\vec{x}=(\Delta x,\Delta y)$),

$$\mathbf{S}=\sum_i|i\rangle\langle i|\otimes|\vec{x}+\Delta\vec{x}_i\rangle\langle\vec{x}|=\mathbf{I}+\underbrace{(\mathbf{I}\otimes\sigma_3)\Delta x\,\partial_x+(\sigma_3\otimes\mathbf{I})\Delta y\,\partial_y}_{\Delta\mathbf{S}}\,,$$
(61)

where $\sigma_i$ are the Pauli matrices. Similarly, we can write the coin operator in the form,

$$\mathbb{C}(\theta_1,\theta_2,\xi,\alpha) = \frac{e^{i(\alpha-\xi)}}{2}\Bigl[\bigl(\mathbb{I}_4+\sigma_3\otimes\sigma_3\bigr)\cos\theta_1 - i\bigl(\sigma_1\otimes\sigma_1-\sigma_2\otimes\sigma_2\bigr)\sin\theta_1\Bigr]$$
$$+\frac{e^{i(\alpha+\xi)}}{2}\Bigl[\bigl(\mathbb{I}_4-\sigma_3\otimes\sigma_3\bigr)\cos\theta_2 - i\bigl(\sigma_1\otimes\sigma_1+\sigma_2\otimes\sigma_2\bigr)\sin\theta_2\Bigr]\,,$$
(62)

where we have introduced another $U(1)$ phase $\alpha$. Using the parameterization,,

$$\theta_1(t,x,y)=\theta_1^{(0)}(t,x,y)+\bar{\theta}_1(t,x,y)\epsilon^\mu\,,\qquad \theta_2(t,x,y)=\theta_2^{(0)}(t,x,y)+\bar{\theta}_2(t,x,y)\epsilon^\nu\,,$$
$$\xi(t,x,y)=\xi^{(0)}(t,x,y)+\bar{\xi}(t,x,y)\epsilon^\rho\,,\qquad \alpha(t,x,y)=\alpha^{(0)}(t,x,y)+\bar{\alpha}(t,x,y)\epsilon^\omega\,,$$
(63)

and that $\mathbf{C}(\theta_1^{(0)},\theta_2^{(0)},\xi^{(0)},\alpha^{(0)})=\mathbf{I}$, translates to,

$$\theta_1^{(0)}=k_1\pi\,,\quad \theta_2^{(0)}=k_2\pi\,,\quad \xi^{(0)}=\pi(n-m)+\frac{k_1-k_2}{2}\pi\,,\quad \alpha^{(0)}=\pi(m+n)-\frac{k_1+k_2}{2}\pi\,,$$
(64)

subject to $(k_1,k_2,n,m)\in\mathbb{Z}$. Most generally, we can take $\Delta t\sim\epsilon^{\mathfrak{T}}$, $\Delta x\sim\Delta y\sim\epsilon^\delta$ *i.e.* all parameters with different scalings. However to pertain to the most simplest scaling, we will take $\mathfrak{T}=\delta=\mu=\nu=\rho=\omega=1$. This also ensures maximal contribution from all parameters. With this scaling, to the leading order (with $\cos\theta=1+O(\theta)^2$ and $\sin\theta\approx\theta$),

$$\mathbf{C}(\bar{\theta}_1,\bar{\theta}_2,\bar{\xi},\bar{\alpha})=\mathbf{I}+i\epsilon\bigl(\bar{\alpha}\mathbf{I}-\bar{\xi}\sigma_3\otimes\sigma_3\bigr)-\frac{i\epsilon}{2}\sigma_1\otimes\sigma_1\Theta_+-\frac{i\epsilon}{2}\sigma_2\otimes\sigma_2\Theta_-\,,$$
(65)

where,$\Theta_\pm = \bar{\theta}_1 e^{i(\alpha-\xi+k_1\pi)}\pm e^{i(\alpha+\xi+k_2\pi)}\bar{\theta}_2$. Putting (61) and (65) in (60) and expanding to $O(\epsilon)$, we can write the perturbative equation of motion,

$$\partial_t\Psi_w=\underbrace{\left((\sigma_3\otimes\mathbf{I})\partial_x+(\mathbf{I}\otimes\sigma_3)\partial_y+i\bigl(\bar{\alpha}\mathbf{I}-\bar{\xi}\sigma_3\otimes\sigma_3\bigr)-\frac{i}{2}\bigl(\sigma_1\otimes\sigma_1\Theta_++\sigma_2\otimes\sigma_2\Theta_-\bigr)\right)}_{Hamiltonian}\Psi_w\,.$$
(66)

Further, defining $X=1/2(x+y)$ and $Y=1/2(x-y)$, the eoms of components $(\mathcal{A}^{(0)},\mathcal{A}^{(3)})$ and $(\mathcal{A}^{(1)},\mathcal{A}^{(2)})$ couple together,

$$\bigl(\partial_t-\partial_X-i(\bar{\alpha}-\bar{\xi})\bigr)\mathcal{A}^{(0)}=-\frac{i}{2}(\Theta_+-\Theta_-)\mathcal{A}^{(3)}\,,$$
$$\bigl(\partial_t+\partial_Y-i(\bar{\alpha}+\bar{\xi})\bigr)\mathcal{A}^{(1)}=-\frac{i}{2}(\Theta_++\Theta_-)\mathcal{A}^{(2)}\,,$$
$$\bigl(\partial_t+\partial_X-i(\bar{\alpha}-\bar{\xi})\bigr)\mathcal{A}^{(3)}=-\frac{i}{2}(\Theta_+-\Theta_-)\mathcal{A}^{(0)}\,,$$
$$\bigl(\partial_t-\partial_Y-i(\bar{\alpha}+\bar{\xi})\bigr)\mathcal{A}^{(2)}=-\frac{i}{2}(\Theta_++\Theta_-)\mathcal{A}^{(1)}\,.$$
(67)

After some rearrangement, (67) can be put in the form of Dirac equation for massive fermions,

$$\left(i\gamma^\mu D_\mu^\pm - \mathcal{M}_\pm\right)\psi_\pm = 0, \tag{68}$$

where $D_\mu^\pm = \partial_\mu - iA_\mu^\pm$ and $A_\mu^\pm = (\bar\alpha \mp \bar\xi, 0)$ and the mass matrix is given by $\mathcal{M}_\pm = (\Theta_+ \mp \Theta_-)/2$. The explicit solutions for the Dirac equation is,

$$\psi_\pm(t,y) = e^{i(\bar\alpha \pm \bar\xi)t}\widetilde{\psi}_\pm(t,y),$$
$$\widetilde{\psi}_\pm(t,y) = \begin{cases} e^{-ip_0 t - ipy}u(p), \\ e^{ip_0 t + ipy}v(p), \end{cases} \qquad p_0^2 = p^2 + m_i^2, \quad p_0 > 0, \tag{69}$$

as the positive and negative energy solutions respectively. Explicitly,

$$u(p) = \begin{pmatrix} Q_- \\ Q_+ \end{pmatrix}, \qquad v(p) = \begin{pmatrix} Q_- \\ -Q_+ \end{pmatrix}. \tag{70}$$

where $Q_\pm = \sqrt{p_0 \pm p}$, $m^2 = (Q_-Q_+)^2$ and $Q_+^2 + Q_-^2 = 2p_0$. Under $p \to -p$, $Q_\pm \to Q_\mp$. The vectors satisfy,

$$u^\dagger(p)u(p) = 2p_0 = v^\dagger(p)v(p), \qquad u^\dagger(p)v(-p) = 0. \tag{71}$$

The explicit solution for each energy (positive or negative frequency) is then $\widetilde{\psi}_\pm(t,y) = e^{-ip_0 t}\widetilde{\psi}_\pm(y)$ where,

$$\widetilde{\psi}_\pm(y) = \int_{-\infty}^\infty \frac{dp}{2\pi}\frac{e^{-ipy}}{\sqrt{2p_0}}\left(a_p^\pm u(p) + b_{-p}^\pm v(-p)\right), \tag{72}$$

where $-p = (p_0, -\mathbf{p})$. The coefficients, $a_\mathbf{p}$ and $b_{-\mathbf{p}}$ can be determined from the inverse transform,

$$a_p^\pm = \int_{-\infty}^\infty dy\, \frac{e^{ipy}}{\sqrt{2p_0}}u^\dagger(p)\widetilde{\psi}_\pm(y), \qquad b_{-p}^\pm = \int_{-\infty}^\infty dy\, \frac{e^{ipy}}{\sqrt{2p_0}}v^\dagger(-p)\widetilde{\psi}_\pm(y). \tag{73}$$

The normalization condition for each time slice $t$ is,

$$\int_{-\infty}^\infty d\mathbf{y}\, \widetilde{\psi}_\pm^\dagger(y)\widetilde{\psi}_\pm(y) = 1 = \int_{-\infty}^\infty \frac{d\mathbf{p}}{2\pi}\left(|a_\mathbf{p}^\pm|^2 + |b_{-\mathbf{p}}^\pm|^2\right). \tag{74}$$

We choose the initial wave function to be gaussian in the spatial coordinates,

$$\widetilde{\psi}_+(0,y) = \left(\frac{2}{\pi\sigma^2}\right)^{1/4}e^{-y^2/\sigma^2}\left(\cos\theta_+/2|\uparrow\rangle + \sin\theta_+/2e^{i\mu_+}|\downarrow\rangle\right), \tag{75}$$

that fixes,

$$a_p^+ = (2\pi\sigma^2)^{1/4}\frac{e^{-p^2\sigma^2/4}}{\sqrt{2p_0}}\left(Q_-\cos\theta_+/2 + Q_+\sin\theta_+/2e^{i\mu_+}\right), \tag{76}$$

$$b_{-p}^+ = (2\pi\sigma^2)^{1/4}\frac{e^{-p^2\sigma^2/4}}{\sqrt{2p_0}}\left(Q_+\cos\theta_+/2 - Q_-\sin\theta_+/2e^{i\mu_+}\right), \tag{77}$$

and similarly for the initial wave function for $\widetilde{\psi}_-(0,y)$. The wave-function for the random walk can be constructed from these fermions, by using the mapping,

$$|\Psi_w(t,\vec{x})\rangle = M \cdot (|\uparrow\rangle \otimes |\psi_+\rangle + |\downarrow\rangle \otimes |\psi_-\rangle), \quad \text{where} \quad M = \begin{pmatrix} 1 & 0 & 0 & 0 \\ 0 & 0 & 1 & 0 \\ 0 & 0 & 0 & 1 \\ 0 & 1 & 0 & 0 \end{pmatrix}. \tag{78}$$

The energy for each fermion is,

$$E_\pm(p) = V_\pm \pm \sqrt{p^2 + m_\pm^2}\,. \tag{79}$$

Specifically for $V_\pm \geq m_\pm$ the lower bound is positive for,

$$E(p) = V_\pm - \sqrt{p^2 + m_\pm^2} \geq 0 \;\rightarrow\; -\sqrt{V_\pm^2 - m_\pm^2} \leq p \leq \sqrt{V_\pm^2 - m_\pm^2}\,. \tag{80}$$

# 7 Conclusions and future directions

A $2d$ DQRW requires a $SU(4)$ coin operator which has 15 parameters. In this work, we have chosen a special coin operator built from Bell pairs and containing only 3 parameters. The relatively simple form of the coin operator renders the DQRW exactly solvable. However, the coin incudes a non-trivial entanglement in the system. In order to probe the entangling properties of the coin operator:

- We compute the entanglement induced by the coin on the spin and position degrees of freedom. This function oscillates with time around a reference constant value.

- We explore the *Entangling Power* of the coin operator which measures its capacity to induce entanglement in the state, starting from an initial tensor product state. Numerically, we compute the Entangling power of the coin as a function of time and for specific choices of the coin parameters.

- We compute the *Generalized Relative Rényi Entropy* functions between the density matrix operators for the initial tensor product state and the final entangled state. We analyze the relative entropies as a function of the coin and time.

Both the measures, behave functionally in the same manner as the entanglement. As a bonus, the continuum limit of the random walk algorithm reduces to two $1d$ massive fermions coupled to synthetic gauge fields. The wave-function for the random walk can be recovered as a non-trivial linear combination of these massive fermions. We conclude the work with discussions and future questions to be addressed.

- We intend to generalize the algorithm using a non-trivial shift operator and/or a feature dependent coin operator. Such generalizations can describe variety of phenomenon such as scattering, tunneling, optimization techniques and so on. An entanglement based random walk approach can also be used to distinguish between topological phases [4] and for the distinction between pure and mixed states using the Entanglement of Purification (EoP) [28].

- The generalized version of entangling power is through *Concurrence matrix* [15, 24] or $n-$tangle operators for higher qubits and higher dimensions [25–27]. It would be interesting to see if the generalizations can be used as an order parameter to determine entanglement evolutions in real systems. An interesting avenue would be to explore the entangling power of mixed states [29, 30].

- An immediate question would be to understand the complexity [31, 32] of the quantum circuit describing the algorithm. This question can be addressed with ease for the random walk algorithm and various approaches of Neilson-complexity [33, 34] and the Krylov-complexity (state-operator complexity) can be compared and extended.

- The structure of the coin operator gives us two one dimensional $1d$ free fermions coupled to synthetic gauge fields. The same structure of the coin in $d$ dimensions, should give us $d$ one dimensional coupled fermions. A feature dependent coin will introduce a non-trivial profile for the gauge potential. If we want to introduce interactions between the fermions, what kind of modification to the coin operator would we need?

## Acknowledgements

The authors thank Siddhartha E. M. Guzmán and Parthiv Halder for collaborating during the initial stages of the work. The authors also thank Aranya Bhattacharya for useful comments and valuable insights on the work.

**Funding information** KS is supported by FAPESP grant 2021/02304-3.

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
