# Peer review of "A simple coin for a $2d$ entangled walk"

_SciPost Physics Core, doi:SciPost Phys. Core 6, 048 (2023)_

## Round 1 · Referee Report · Anonymous (Referee 1) · 2023-5-24

Strengths

Closed-form solution for a problem of wide applicability, which may well simplify future calculations.

Weaknesses

The scope of the paper is somewhat limited, it remains to be seen if indeed this closed form solution is practical.

Report

The manuscript presents a closed-form solution, in terms of a special function, of a problem of wide applicability: The quantum walk in discrete time on a two-dimensional lattice. The closed-form solution of the 1D problem is known, I have not found the 2D case in the literature. So I do think that this is a valuable contribution to the literature. The topic is timely (in particular, with applications in the context of quantum search algorithms). I find this work suitable for publication in SciPost Physics Core.
  • validity: high
  • significance: good
  • originality: good
  • clarity: high
  • formatting: good
  • grammar: good

Author:  Kallol Sen  on 2023-05-25  [id 3683]

(in reply to Report 1 on 2023-05-24)
Category:
remark

The authors thank the reviewer for insightful comments on the work. The practical relevance of the exact solutions in 2d lies in the context of the algorithm. We have focussed on the question of finding a coin operator that induces entanglement in the 2d walk. A coin built from minimal set of bell pairs admits analytic solution while preserving the versatile dynamical aspects of the walk. This is precisely the context in which the analytical solutions become appealing. This serves as a building block for constructing more generalized walk algorithms in higher dimensions and for quantum searches. Also, with analytical solutions, essential dynamical features of the walk e.g. entanglement, entropy and entangling power are extrapolated for asymptotic expansions leading to interesting continuum limits connecting quantum field theories with real time simulations of their discrete versions.

---

## Editorial Decision

published